# Medical cost of acute diarrhea in children in ambulatory care

**Xavier Sánchez**[1☉], **Gerardine Leal**[2‡], **Angel Padilla**[2‡], **Ruth Jimbo**[1☉¤*]

1 Centro de Investigación en Salud para América Latina (CISeAL), Facultad de Medicina, Pontificia Universidad Católica del Ecuador (PUCE), Quito, Ecuador, 2 Postgrado de Medicina Familiar y Comunitaria, Pontificia Universidad Católica del Ecuador (PUCE), Quito, Ecuador

☉ These authors contributed equally to this work.
¤ Current address: Centro de Investigación para la Salud en América Latina (CISeAL), Pontificia Universidad Católica del Ecuador (PUCE), Quito, Ecuador
‡ GL and AP also contributed equally to this work.
* rejimbo@puce.edu.ec

**Data Availability Statement:** All relevant data are within the paper and its Supporting Information files.

**Funding:** The authors received no specific funding for this work.

## Abstract

### Objective

The aim of this study was to estimate the direct medical cost per episode and the annual cost for acute diarrhea (AD) in children under five years of age in Ambulatory Care Centers of the Ministry of Public Health (MOPH) of Ecuador.

### Methods

A cost of illness study with a provider perspective was carried out through a micro-costing of health resources and valuated in international dollars. Medical consultations and laboratory tests were valued using the tariff framework of services for the National Health System and for the prescribed medications, a reported cost registry of pharmacy purchases made in the year of study was used.

### Results

A total of 332 electronic health records of children under five years of age were included in the analysis. Laboratory tests were performed on 37.95% (126/332), medications were pre-scribed to 93.67% (311/332) of the children, and antimicrobials were prescribed to 37.35% (124/332) of the children, representing an antibiotic prescription rate of 26.51% (88/332) and an antiparasitic prescription rate of 10.84% (36/332). The mean cost of the MOPH per child per episode of AD was US$45.24 (2019 dollars) (95% CI:43.71 to 46.76).

### Conclusion

The total estimated cost of AD in children under five years of age for the MOPH in 2019 was about US$6,645,167.88 million (2019 dollars) (95% CI: 6,420,430.77 to 6,868,436.12). A high proportion of the direct medical cost of AD in children under five years of age in outpatient settings is due to unnecessary laboratory tests.

**Competing interests:** The authors have declared that no competing interests exist.

## Introduction

Diarrheal disease is one of the leading causes of death in children in developing countries [1]. Acute diarrhea (AD) is defined as the passage of three or more loose or liquid stools per day [2]. It primarily occurs in children during the first five years after birth, and particularly in the second half-year and in little children [3]. It is estimated that 2.5 billion cases of diarrheal disease occur each year in children under five years of age, and an average of more than 1,400 children die each day [4]. Evidence shows that AD is common in places with poor access to health care, safe water and sanitation, which is often observed in low and middle-income countries [5].

The most common cause of AD is gastrointestinal infections caused by viruses, bacteria and occasionally parasite. Rotavirus is responsible for 60–70% of all diarrheal diseases; however, *E. coli*, *Campylobacter*, *Yersinia* and *Salmonella spp.* are common bacterial causes of AD [6]. The diagnosis of AD is mainly based on the medical interview, which provides accurate information about the duration of the diarrhea and the characteristics of the accompanying symptoms, such as abdominal pain, vomiting, and fever. Healthy children without any comorbidity do not require specific diagnosis tests in cases of AD since the results of laboratory and microbiological tests do not change the treatment of the episode; so, testing should be kept to a minimum and reserved for patients with severe illness [7].

Most episodes of AD are self-limiting, and prevention of dehydration is the main goal of diarrhea treatment. Medications such as antibiotics, antiparasitic drugs and antidiarrheal agents are not necessary and can be harmful for infants or children with diarrhea [8, 9]. On occasion, antibiotics may be used in cases of bacterial infection when a specific cause of the diarrhea has been found or is strongly suspected, particularly after recent travel [10]. However, evidence shows that the most commonly reported inappropriate measures used in AD treatment are the unwarranted prescription of antimicrobial, antidiarrheal and antimotility drugs, especially in low- and middle-income countries [11].

Unnecessary use of resources in the treatment of AD leads to inefficient use of scarce healthcare resources. It is estimated that a significant proportion of medical tests and drug prescriptions is unnecessary around the world; this is of particular concern because of the potential financial effect of excessive resource utilization on healthcare systems [12]. To date, no research has attempted to estimate this cost burden on Ecuador's healthcare system, even when AD is a leading cause of morbidity in children [13]. Demonstrating the importance of this problem to policy makers and clinicians would provide information to aid budgetary decision making and understanding of the need for cost containment interventions.

## Materials and methods

Ethics approvals for the protocol and the study were granted by the Subcommittee for Research Ethics on Human Beings–Pontificia Universidad Católica del Ecuador with authorization code SB-CEISH-POS-749. The Ethics Committee established that there was no need for informed consent for this study.

## Objective

The aim of this study was to estimate the direct cost per episode and the annual cost of AD in the Ambulatory Care Centers of the Ministry of Public Health (MOPH) care system in 2019. Participants included children under five years of age, who had been diagnosed with AD.

## Setting

Ecuador's health system is made up of two sectors, public and private. The MOPH is the governing entity of health in the country, offering health care services to the entire Ecuadorian population. Its financing comes from state contributions, and within the public health system it has a coverage of about 62%. The rest of the population is covered by a public subsystem such as social security institutions like the Ecuadorian Institute of Social Security (IESS), The Institute of Security of the Armed Forces (ISSFA) and the Institute of Social Security of the National Police (ISSPOL), or the private system in the case of segment of the population that has the means to pay [14, 15].

District 17D03 of the city of Quito belongs to the public health network of Ecuador. This district has a total assigned population of 430,000 inhabitants and almost 36% is a pediatric population (0–18 years). The district is distributed in 21 ambulatory care centers; these health centers are distributed in rural and urban areas of the city and offer health services to 85% of the population [15, 16].

## Sample

A simple probabilistic type sampling was performed. To calculate the sample, the total number of cases registered by AD in the pediatric population under five reported by the MOPH in the district in 2019 was used, in S1 Table. We considered an acute diarrhea diagnosis to be any diagnosis registered according to the ICD-10 classification, as the following: **A00:** Cholera, **A01:** Typhoid and paratyphoid fevers, **A02:** Other salmonella infections, **A03:** Shigellosis, **A04:** Other bacterial intestinal infections, **A05:** Other bacterial foodborne intoxication not elsewhere classified, **A06:** Amoebiasis, **A07:** Other protozoal intestinal diseases, **A08:** Viral and other specified intestinal infections and **A09:** Other gastroenteritis and colitis of infectious and unspecified origin.

The following formula was applied to calculate the sample for a finite universe:
$$n = N^*Z^{2*}p^*q \,/\, d^{2*}(N\text{-}1) + Z^{2*}p^*q.$$

where $N$ is the population size, $Z$ is the confidence level (95%), $p$ is the probability of success, or expected proportion (50%), $q$ is the probability of failure (50%), and $d$ is precision (5% of maximum admissible error in terms of proportion). The subsequently studied sample comprised 332 Electronic Health Records (EHR).

**Identification of resource use and data collection.** *Perspective.* A provider perspective was used, which included resources provided by the MOPH during the period between the initial consultation and resolution of the illness up to a maximum of five weeks. This included medical care, laboratory tests, and medication as primary resources.

*Data source.* The data source for the resources used and data collect in this study was the Electronic Health Records (EHR) of the patients from 21 health centers of the Ministry of Public Health in the District 17D03 in Ecuador during 2019 with AD as a primary diagnosis. This district has been using the EHR since 2010. Physicians enter information directly into the EHR on a computer during the outpatient appointment. Information was manually extracted from the EHR by two reviewers. The tests ordered and medication administered were verified against laboratory and pharmacy records.

## Valuation of resources

All resources were valued in dollars at 2019 prices. To make the monetary values internationally comparable, they were converted to international dollars (a hypothetical currency with the same purchasing power of goods and services in all countries) using the purchasing power parity (PPP) conversion factor for Ecuador [17]. The cost of the resources was calculated based on

information from official national cost sources. The National Health System service fee schedule and pharmacy drug purchase records were used [18]. Unit costs are available in S2 Table.

### Estimating the annual cost for the MOPH

To estimate the annual cost of AD in 2019 for the MOPH in children under five years of age, the estimated cost per episode was combined with the total number of cases of AD treated by the MOPH at the first level of health for that population. The annual cost to the population was calculated as follows: *Annual cost 2019 = (cost per episode per child in 2019) × (total number of episodes in 2019)*. The number of episodes in 2019 is the number of children under five years of age who were treated with AD in a primary health center of the MOPH in 2019, according to ICD-10 codes taken from the official of national statistics [13]. All analyses were carried out using SPSS v25 software.

## Results

### Participant characteristics

332 eligible EHR were randomly selected and proofed for suitability for inclusion in the cost-of-illness study. The general characteristics of our sample are described in Table 1.

In the same of children, the male gender was more frequent with 57.53% (191/332) and the mean age was 1.29 years. Most of the registered children, 89.75% (198/332), had no comorbidities; chronic malnutrition and anemia were the most frequent comorbidities reported, with 3.92% (13/332) and 3.31% (11/332) respectively. Of the total number of AD consultations, general practitioner consultations were more frequent with 45.78% (152/332) followed by pediatrician consultations with 25.30% (84/332).

A total of 91 healthcare professionals treated all the cases coded as AD, with females being the most frequent (64.84% vs. 35.16%) and the mean age of the professionals was 40.5. Two types of health care professionals were included: general physicians (Rural Doctor and General Practitioner) and specialists (Family Medicine Physician and Pediatrician), general physicians were the most frequent health professionals (68.12% vs. 31.88%).

The ICD-10 diagnostic code most frequently assigned by physicians was A09 (Other gastroenteritis and colitis of infectious and unspecified origin) in 98.19% (326/332) of the cases (Table 2).

### Resource use

The total resource use in the sample by item is shown in Table 3. A total of 1546 resources were used in 332 children. All children had one primary care contact, and the re-consultation rate was 9.94% (33/332); out of these, just one child needed two re-consultations.

Laboratory tests were performed in 37.95% (126/332) children, representing 320 units of laboratory tests dispensed, of which 38.44% (123/320) were coprological tests, 25% (80/320) fecal leukocyte stain (WBC, stool), 11.25% (36/320) rotavirus tests, 11.88% (38/320) blood count, and 6.88% (22/320) urinalysis (Table 3); in some children, more than one test was ordered, the most frequent combination of tests were coprological test/fecal leukocyte stain in 63.49% (80/126) and coprological test/blood count at 35% (35/126).

Medications were prescribed to 93.67% (311/332) of children. Oral rehydration salts were the most prescribed medication, with 77.81% (242/311), followed by anti-inflammatory drugs with 43.41% (135/311). Antibiotics were prescribed to 88 children, representing an antibiotic prescription rate of 26.51% (88/332) and antiparasitic were prescribed to 36 children, indicating a prescription rate of 10.84% (36/332). Zinc sulfate was prescribed to only 3.61% (12/332) (Table 3).

**Table 1. Characteristics of the sample.**

| Variable | n (%) |
|---|---|
| **Total electronic health records** | **332 (100)** |
| **Gender of patients** | |
| Female | 141 (42.46) |
| Male | 191 (57.53) |
| **Age of patients** | |
| Mean (SD) | 1.85 (1.29) |
| **Comorbidity** | |
| No comorbidity | 298 (89.76) |
| Chronic malnutrition | 13 (3.92) |
| Anemia | 11 (3.31) |
| Asthma | 2 (0.60) |
| Obesity | 2 (0.60) |
| Autism | 1 (0.30) |
| Clift lip | 1 (0.30) |
| Down's syndrome | 1 (0.30) |
| Epilepsy | 1 (0.30) |
| Microcephaly | 1 (0.30) |
| Prematurity | 1 (0.30) |
| **Consultations by type of healthcare professional** | |
| Rural Doctor consultations | 57 (17.16) |
| General Practitioner consultations | 152 (45.78) |
| Family Medicine Physician consultations | 39 (11.74) |
| Pediatrician consultations | 84 (25.30) |
| **Total healthcare professionals** | **91 (100)** |
| **Gender of healthcare professionals** | |
| Female | 59 (64.84) |
| Male | 32 (35.16) |
| **Age of healthcare professionals** | |
| Mean (SD) | 40.51 (13.41) |
| **Types of healthcare professionals** | |
| Rural Doctor | 5 (5.49) |
| General Practitioner | 57 (62.63) |
| Family Medicine Physician | 26 (28.57) |
| Pediatrician | 3 (3.29) |

SD, standard deviation.

A total of 860 units of medication were dispensed, of these oral rehydration salts represented 60.30% (553/860) of the units used, anti-inflammatory drugs 17.79% (136/860), antibiotics 11.16% (96/860), antiparasitic 6.63% (57/860), and zinc sulfate 2.09% (18/860) (Table 3).

Medical contact accounts for 23.67% (366/1546) of the total resource use, prescriptions account for 55.63% (860/1546) and laboratory tests account for 20.70% (320/1546) (Table 4).

## Cost per episode and overall cost

The mean cost per child per resource used is displayed in Table 3. The mean cost to the MOPH per child per episode of AD was US$45.24 (2019 dollars) (95% CI:43.71, 46.76), the mean cost of laboratory tests was US$15.98 (95% CI: 14.20, 17.76) (2019 dollars), and the mean cost of drugs prescriptions was US$2.51 (95% CI: 2.27, 2.76) (2019 dollars).

**Table 2. Consultations of acute diarrhea by type of healthcare professional.**

| Health professional | ICD-10 diagnosis n (%) | | | | Total |
|---|---|---|---|---|---|
| | A06.9 | A07.1 | A08.4 | A09 | |
| Rural Doctor | 0 (0) | 0 (0) | 0 (0) | 57 (17.49) | 57 |
| General Practitioner | 1 (25) | 1 (100) | 0 (0) | 150 (46.01) | 152 |
| Family Medicine Physician | 0 (0) | 0 (0) | 0 (0) | 39 (11.96) | 39 |
| Pediatrician | 3 (75) | 0 (0) | 1 (100) | 80 (24.54) | 84 |
| **Total** | **4 (100)** | **1 (100)** | **1 (100)** | **326 (100)** | **332** |

A06.9, Amoebiasis, unspecified; A07.1, Giardiasis [lambliasis]; A08.4, Viral intestinal infection, unspecified; A09, Other gastroenteritis and colitis of infectious and unspecified origin.

The overall cost for the whole sample is presented in Table 4. The total cost of AD for the whole sample was US$15,008.15 (95% CI: 14,511.72, 15,524.32) (2019 dollars). The first consultation accounted for 76.38% of the total cost of the sample and the subsequent consultations accounted for 5.03%. Laboratory tests accounted for 13.38% (US$2,007.74) (2019 dollars) and prescriptions accounted for 5.21% (US$782.14) (2019 dollars) of the overall cost.

The cost by type of laboratory test is shown in Table 5. Rotavirus tests accounted for 38.32% of the overall cost of laboratory tests, follow by coprological test with 32.41%, and white blood cells in stool test with 10.16%.

The cost by type of prescription is reflected in Table 6. Oral rehydration salts accounted for 38.82% of the overall cost of prescriptions, followed by any type of antimicrobial drug (antibiotic/antiparasitic) with 28.13%, analgesics/anti-inflammatory drugs with 22.73%, and zinc sulfate with 10.29%.

**Table 3. Resources used and mean cost per child.**

| Direct sanitary resources | Number of patients | Units dispensed | Mean Cost per child (95%CI)[a] |
|---|---|---|---|
| **Consultations** | | | |
| First consultations | 332 | 332 | 34.49 (34.40–34.55) |
| Re-consultations | 33 | 34 | 22.88 (21.53–24.24) |
| **Laboratory tests** | | | |
| Blood count | 38 | 38 | 5.10 (5.10–5.10) |
| Urinalysis | 22 | 22 | 5.69 (5.69–5.69) |
| Gram stain & fresh drop | 3 | 3 | 2.75 (2.75–2.75) |
| Coprological test | 123 | 123 | 5.29 (5.29–5.29) |
| WBC, stool | 80 | 80 | 2.55 (2.55–2.55) |
| Rotavirus | 36 | 36 | 21.37 (21.37–21.37) |
| FOBT | 18 | 18 | 3.14 (3.14–3.14) |
| **Prescriptions** | | | |
| Antibiotics | 88 | 96 | 1.75 (1.27–2.22) |
| Antiparasitic | 36 | 57 | 1.84 (1.65–2.02) |
| Anti-inflammatory | 135 | 136 | 1.31 (1.25–1.37) |
| Oral rehydration salts | 242 | 553 | 1.25 (1.18–1.33) |
| Zinc Sulfate | 12 | 18 | 6.10 (4.59–7.61) |
| **Total cost** | **332** | **1546** | **45.24 (43.71–46.76)** |

CI, Confident interval; WBC, Fecal leukocyte stain; FOBT, Fecal occult blood test.
[a]Cost presented in International Dollars 2019.

**Table 4. Total resource use and overall cost of direct sanitary resources.**

| Direct sanitary resources | Number of patients | Units dispensed | % Resource use | Overall Cost[a] | % Overall Cost[a] |
|---|---|---|---|---|---|
| First consultations | 332 | 332 | 21.47 | 1,1462.98 | 76.38 |
| Re-consultations | 33 | 34 | 2.20 | 755.29 | 5.03 |
| Laboratory tests | 126 | 320 | 20.70 | 2,007.74 | 13.38 |
| Prescriptions | 311 | 860 | 55.63 | 782.14 | 5.21 |
| **Total cost** | **332** | **1546** | **100.00** | **15,008.15** | **100.00** |

[a]Costs presented in International Dollars 2019.

## Annual cost to MOPH

We combined the cost per episode results with data on prevalence. The number of consultations for AD in children under five years of age in the ambulatory care centers of the MOPH, according to the National Direction of Statistics and Analysis of the Health Information (DNEAIS), in 2019 was 146,887 [19]. Thus, the annual cost to the MOPH was at least US $6,645,167.88 (2019 dollars) (95% CI: 6,420,430.77 to 6,868,436.12).

## Discussion

To the best of our knowledge, this is the first study to measure AD management costs in children under five years of age in Ecuador. Our data provide statistics on resource use and costs from a payer perspective. The cost was estimated from medical records with diagnoses coded according to ICD-10, applying a micro-cost methodology by involving the direct identification and valuation of the resources consumed treating each patient, which improves the accuracy of cost estimation [20].

Medical appointments represent 23.67% of the resources used per AD episode. All children had at least an initial consultation and the rate of re-consultation was almost 10%, which is less than that reported in the literature [21]. Laboratory tests were ordered in almost 38% of the children, accounting for around the 20% of the resources used.

Medication prescription was the most frequently used resource in the treatment of AD cases. It accounted for 55.63% of all resource units administered. Oral rehydration salts, anti-inflammatory drugs and antibiotics were the most prescribed drugs, 72.89%, 40.66%, and 26.51% respectively; these results are similar to other studies [22–24]. Regardless of this, it is important to mention that the use of resources in medical care may vary according to their availability.

**Table 5. Cost by type of laboratory test.**

| Type of laboratory test | Units | Total Cost[a] | % |
|---|---|---|---|
| Blood count | 38 | 193.8 | 9.65 |
| Urinalysis | 22 | 125.18 | 6.23 |
| Gram stain & fresh drop | 3 | 8.25 | 0.41 |
| Coprological test | 123 | 650.67 | 32.41 |
| WBC, stool test | 80 | 204 | 10.16 |
| Rotavirus | 36 | 769.32 | 38.32 |
| FOBT | 18 | 56.52 | 2.82 |
| **Total** | **320** | **2,007.74** | **100** |

WBC, stool: Fecal leukocyte stain; FOBT, Fecal occult blood test.

[a]Costs presented in International Dollars 2019.

**Table 6. Cost by type of medication.**

| Type of medication | Units | Total Cost[a] | % |
|---|---|---|---|
| Amoxicillin | 14 | 59.24 | 7.57 |
| Azithromycin | 9 | 20.94 | 2.68 |
| Cephalexin | 3 | 7.76 | 0.99 |
| Clarithromycin | 5 | 34.31 | 4.39 |
| Trimethoprim/Sulfamethoxazole | 65 | 31.86 | 4.07 |
| Metronidazole | 32 | 55.84 | 7.14 |
| Albendazole | 25 | 10.29 | 1.32 |
| Ibuprofen | 9 | 21.43 | 2.74 |
| Paracetamol | 127 | 156.37 | 19.99 |
| Oral rehydration salts | 553 | 303.63 | 38.82 |
| Zinc Sulfate | 18 | 80.47 | 10.29 |
| **Total** | **860** | **782.14** | **100** |

[a]Costs presented in International Dollars 2019.

Numerous studies have reported the economic burden of childhood diarrhea in different settings; however, empirical data on the cost of diarrheal illness is sparse [25–29]. We found that the average cost of AD per episode in children under five years of age in Ecuador was US $45.24 (2019 dollars). Our results are similar to those of some countries reported by Rheingans *et al.* [30] for Latin American and Caribbean countries, where total direct medical cost in outpatient settings was US$18.45 in Argentina, US$15.03 in Brazil, 47.25 US$ in Chile, US$30.26 in Honduras, US$33.58 in Mexico, US$39.50 in Panamá and US$30.96 in Venezuela (monetary change to 2019). Nevertheless, the proportion of laboratory and medication costs were higher in Rheingans' study, between 50% and 80%, compared to our data, which was less than 20%. Alternatively, a systematic review of the cost of childhood diarrhea in 137 low- and middle-income countries (LMICs) estimated an average (weighted) US$52.16 per outpatient episode of AD [31], and the direct medical cost represented almost 80% of the total cost, US $41.21 (2020 dollars). These findings are more comparable with our results.

Considerable variation is found in the cost estimates reported in the literature, variation that can be attributed to methodological differences between studies, in particular variation in the perspective adopted by the studies and the categorization of the costs [32–34]. Additionally, the direct cost associated with treatment could vary according to the capabilities and resources available in different health units, even within the same country, leading to some heterogeneity in the estimated costs of the studies.

The results from our study on the cost of illness may be conservative. As part of the sensitivity analysis, the effect of the proportion of laboratory tests was explored. Otherwise healthy children with AD in ambulatory settings do not require a specific diagnostic work-up, as the results of laboratory investigations are unlikely to change their management [10, 35]. According to the Integrated Management of Childhood Illness (IMCI) [36], among other components, to improve the performance of health personnel caring for children under five, it is not necessary to request laboratory tests in outpatient care. However, we found a proportion of laboratory tests ordered of around 13%. If we consider that physicians comply with the recommendations for care according to IMCI, the estimated population cost would be US$ 800,000 lower. Other variables in the study do not greatly affect the results as we found them to be relatively low cost.

Cost-of-illness studies can estimate cost based on epidemiological data, through incidence-based or prevalence-based approaches [37]; the latter is useful for identifying points for cost containment, as this approach provides information on each cost component [38, 39].

The total estimated cost of AD in children under five years of age for MOPH in 2019 was about US$6,645,167.88 (2019 dollars) (95% CI: 6,420,430.77 to 6,868,436.12). If laboratory tests had been minimized and antibiotic use cut by at least half, our annual cost estimate would have been reduced by $1 million. Therefore, an intervention that promotes rational use of these resources would be cost saving.

The Integrated Management of Childhood Illness (IMCI) strategy was designed in 1996 by the World Health Organization (WHO) and the United Nations Children's Fund (UNICEF) to improve the quality of care for children in primary care services and was adopted by Ecuador [40]. According to this strategy, in patients under five years of age with AD, it is not necessary to request laboratory tests, and the use of antibiotics is justified in few circumstances; therefore, our findings show that there may be an overuse of these resources and lack of adherence to IMCI recommendations, a situation that has been reported in other studies [41, 42]. However, the evaluation of the strategy and its implementation is beyond our research objectives.

A limitation of this study is that it focuses only on the population subsidized by the MOPH; although it covers most of the country's population [43], the results may not be generalizable to other public health subsystems. Another limitation is the perspective used in the study; we did not include patient time costs or other associated resource costs. This perspective is narrower and does not include patients' out-of-pocket costs as they are not borne by payers; therefore, the cost of illness may be underestimated. A strength of the study is that the AD cases were selected from the medical records of different health centers, according to ICD-10, which allowed us to obtain data according to their own diagnoses and thus determine the use of resources for such care. Moreover, the study shows relevant data for decision-making stakeholders to consider cost containment measures for the efficient use of medical resources.

## Conclusion

A high proportion of the direct medical cost of AD in children under five years of age in outpatient settings is due to the unnecessary use of laboratory tests. A broader perspective in cost-of-illness studies, including indirect and non-medical costs, could help to better estimate the economic burden of the disease in Ecuador. Adherence to national recommendations for treating AD in children needs to be assessed to identify possible overused resources in primary care.

## Supporting information

**S1 Table. Cases registered as acute diarrhea in the pediatric population under five years of age.**
(DOCX)

**S2 Table. Source and value of unit costs.**
(DOCX)

## Author Contributions

**Conceptualization:** Xavier Sánchez, Ruth Jimbo.

**Data curation:** Xavier Sánchez, Gerardine Leal, Angel Padilla.

**Formal analysis:** Xavier Sánchez, Gerardine Leal, Angel Padilla, Ruth Jimbo.

**Investigation:** Xavier Sánchez, Gerardine Leal, Angel Padilla, Ruth Jimbo.

**Methodology:** Xavier Sánchez, Ruth Jimbo.

**Project administration:** Xavier Sánchez.

**Supervision:** Xavier Sánchez.

**Validation:** Xavier Sánchez, Ruth Jimbo.

**Writing – original draft:** Xavier Sánchez, Ruth Jimbo.

**Writing – review & editing:** Xavier Sánchez, Ruth Jimbo.

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
