## [Decision Letter · Decision Letter 0]

27 Jul 2022

PONE-D-22-19266Medical Cost of Acute Diarrhea in Children in Ambulatory CarePLOS ONE

Dear Dr. Jimbo,

Thank you for submitting your manuscript to PLOS ONE. After careful consideration, we feel that it has merit but does not fully meet PLOS ONE’s publication criteria as it currently stands. Therefore, we invite you to submit a revised version of the manuscript that addresses the points raised during the review process.

We look forward to receiving your revised manuscript.

Kind regards,

Nasser Hadal Alotaibi

Academic Editor

PLOS ONE

Journal Requirements:

Reviewers' comments:

Reviewer's Responses to Questions

**Comments to the Author**

1. Is the manuscript technically sound, and do the data support the conclusions?

Reviewer #1: Partly

Reviewer #2: Partly

2. Has the statistical analysis been performed appropriately and rigorously? 

Reviewer #1: No

Reviewer #2: I Don't Know

3. Have the authors made all data underlying the findings in their manuscript fully available?

Reviewer #1: No

Reviewer #2: Yes

4. Is the manuscript presented in an intelligible fashion and written in standard English?

Reviewer #1: Yes

Reviewer #2: Yes

5. Review Comments to the Author

Reviewer #1: 1. The whole manuscript should be revised for improved English and grammatical mistakes.

2. Line 38, %% should be corrected to %, similarly E.coli should be written as E.Coli. All bacterial pathogens name should be written in italic form.

3. Line 79, ‘The district is distributed in 21 ambulatory medical care centres, These health centres are distributed in rural and urban areas of the city and offer health services to 85% of the population’. According to WHO criteria, 30 prescription per facility should be analysed. So the sample size should be at least 630.

4. Line 123, Estimation of annual cost is not clear. How total number of AD episodes per year of the same patient was calculated?

5. Name of health facilities should be mentioned in a tabular form, mentioning the number of samples taken per facility.

6. I could not find the standard treatment of AD given by WHO. Please correlate your study with STGs of AD with WHO guidelines. You can add another variable of Adherence to STGs in your study. For your ease please read and cite the following article.

https://dx.doi.org/10.21608/bfsa.2020.93568

7. et al should be italic.

8. Conclusion should be rewritten. As lab test is necessary so I don’t think that treatment cost is increased due to unnecessary testing. So revise conclusion according to prescribed medicines and its cost based on local or multinational brands.

9. No statistical test was applied on the variables.

Other comments:

The article focuses on treatment cost which is divided into two types, 1. Medication cost and 2. Lab test cost. On the other hand the article describes number of units dispensed from the facilities. So these two parameters should be clearly separated by headings and discussed accordingly in separate results heading.

Reviewer #2: Authors should be more clear on the methodology and presentation of results in approach that can be understood by a layman. The comments provided in the attached pdf documents should be addressed clearly and correctly.

6. PLOS authors have the option to publish the peer review history of their article (what does this mean?). If published, this will include your full peer review and any attached files.

Reviewer #1: No

Reviewer #2: **Yes: **Oluwasola Stephen Ayosanmi

---

## [Author Response · Author response to Decision Letter 0]

30 Aug 2022

Academic Editor

“1. Please ensure that your manuscript meets PLOS ONE's style requirements, including those for file naming. The PLOS ONE style templates can be found at 

https://journals.plos.org/plosone/s/file?id=ba62/PLOSOne_formatting_sample_title_authors_affiliations.pdf”

Corrections were made to meet PLOS ONE’s style requirements.

“2. Please update your submission to use the PLOS LaTeX template. The template and more information on our requirements for LaTeX submissions can be found at http://journals.plos.org/plosone/s/latex.”

PLOS template was used in this submission.

“3. Please include your full ethics statement in the ‘Methods’ section of your manuscript file. In your statement, please include the full name of the IRB or ethics committee who approved or waived your study, as well as whether or not you obtained informed written or verbal consent. If consent was waived for your study, please include this information in your statement as well.”

Full ethics statement was included in methods section. Lines 78-81

Reviewer #1:

“1. The whole manuscript should be revised for improved English and grammatical mistakes.”

Suggestion accepted. The manuscript has been proofread by a native speaker.

“2. Line 38, %% should be corrected to %, similarly E.coli should be written as E.Coli. All bacterial pathogens name should be written in italic form.”

Suggestion accepted. Correction was made. Lines 52-53

“3. Line 79, ‘The district is distributed in 21 ambulatory medical care centres, These health centres are distributed in rural and urban areas of the city and offer health services to 85% of the population’. According to WHO criteria, 30 prescription per facility should be analysed. So the sample size should be at least 630.”

Observational burden of illness studies are widely used to characterize treatment patterns, resource utilization and costs associated with a disease. For a burden of illness study the aim of sample size calculation is to ensure sufficient precision in descriptive outcomes, e.g. characterized by the width of 95% confidence intervals (CIs). We followed sampling guidelines in cost-of-illness studies designed for practical application in real-world studies (1,2). The sampling approach used in our study complies with the methodological recommendations according to the guidelines for conducting health economics studies.

1. Johnston KM, Lakzadeh P, Donato BMK, Szabo SM. Methods of sample size calculation in descriptive retrospective burden of illness studies. BMC Med Res Methodol [Internet]. 2019;19(1):9. Available from: http://www.ncbi.nlm.nih.gov/pubmed/30626343

2. 1. Jo C. Cost-of-illness studies: concepts, scopes, and methods. Clin Mol Hepatol [Internet]. 2014;20(4):327. Available from: http://e-cmh.org/journal/view.php?doi=10.3350/cmh.2014.20.4.327

“4. Line 123, Estimation of annual cost is not clear. How total number of AD episodes per year of the same patient was calculated?”

The annual cost was estimated as cost per episode per child in 2019 × total number of episodes in 2019. As this is a cost-of-illness study, all costs are imputed to each child to estimate an average cost. If the same child has had several episodes of the disease, this is already considered in the estimate. The methodology (1,2,3) for estimating disease costs provides an estimate of the burden resulting from the prevalence of the disease over a given period, most often a year. This is called the prevalence approach.

1. Hodgson TA, Meiners MR. Cost-of-Illness Methodology: A Guide to Current Practices and Procedures. Milbank Mem Fund Q Health Soc. 1982;60(3):429. Available from: https://www.jstor.org/stable/3349801?origin=crossref

2. Drummond MF: Methods for the economic evaluation of health care programmes. 3rd edition. Oxford: Oxford University Press; 2005.

3. Jo C. Cost-of-illness studies: concepts, scopes, and methods. Clin Mol Hepatol [Internet]. 2014;20(4):327. Available from: http://e-cmh.org/journal/view.php?doi=10.3350/cmh.2014.20.4.327

“5. Name of health facilities should be mentioned in a tabular form, mentioning the number of samples taken per facility.”

Suggestion accepted. Data is displayed in a Supplementary Table S1.

“6. I could not find the standard treatment of AD given by WHO. Please correlate your study with STGs of AD with WHO guidelines. You can add another variable of Adherence to STGs in your study. For your ease please read and cite the following article.

https://dx.doi.org/10.21608/bfsa.2020.93568”

The standard treatment of acute diarrhea is cited. A cost-of-illness study, such as this one, focuses on measuring the amount of resources used for a given disease and estimating the monetary expenditure. Therefore, a variable such as "adherence to standard treatment regimen" is not necessary (it is not a resource used). The implications derived from this variable go beyond our primary objective.

“7. et al should be italic.”

Suggestion accepted. 

“8. Conclusion should be rewritten. As lab test is necessary so I don’t think that treatment cost is increased due to unnecessary testing. So revise conclusion according to prescribed medicines and its cost based on local or multinational brands.”

We politely disagree with the reviewer. According to the guidelines for treating acute diarrhea in children, laboratory testing is not necessary in the outpatient setting, as it will not modify the treatment. Therefore, the use of this resource reflects an unnecessary cost. 

“9. No statistical test was applied on the variables.”

A cost-of-illness study, such as this one, focuses on measuring the amount of resources used for a given disease and estimating the monetary expenditure. Advanced statistical tests are not necessary.

“Other comments:

The article focuses on treatment cost which is divided into two types, 1. Medication cost and 2. Lab test cost. On the other hand the article describes number of units dispensed from the facilities. So these two parameters should be clearly separated by headings and discussed accordingly in separate results heading.”

Suggestion accepted. Results were better discussed following the recommendation but meeting PLOS ONE’s style requirements

Reviewer #2: 

“Authors should be more clear on the methodology and presentation of results in approach that can be understood by a layman. The comments provided in the attached pdf documents should be addressed clearly and correctly.”

Suggestion accepted. This manuscript version has been corrected and rewritten to facilitate comprehension. 

All comments provided were properly addressed. 

1. A definition of international dollar has been added. Lines 130-131.

2. Table 1 now includes information on comorbidities found in the sample. Line 152

3. Table 3 was reworked to improve comprehension. Line 176

4. US$ is used in the manuscript as the standard unit.

“This part is not clear and confusing. The prevalence rate is not the same as the sample size. I disagree that you can extrapolate the data on national prevalence to estimate annual cost. The annual cost should stay within your sample size. This is because health cost will vary by location. So, you can not assume the cost in your location is the same as elsewhere.”

Cost-of-illness studies can be described as prevalence-based or incidence-based approaches, depending on how the epidemiological data are used. Prevalence-based studies can use this procedure to extrapolate cost and resource data over a period of time (1,2). This is called a prevalence approach. We partially agree with the reviewer, as this type of estimation may have some limitations; however, this is discussed in lines 260-655. For this reason, it is advisable to perform a sensitivity analysis (lines 266-276).

1. Hodgson TA, Meiners MR. Cost-of-Illness Methodology: A Guide to Current Practices and Procedures. Milbank Mem Fund Q Health Soc. 1982;60(3):429. Available from: https://www.jstor.org/stable/3349801?origin=crossref

2. Jo C. Cost-of-illness studies: concepts, scopes, and methods. Clin Mol Hepatol [Internet]. 2014;20(4):327. Available from: http://e-cmh.org/journal/view.php?doi=10.3350/cmh.2014.20.4.327

“The term sensitivity analysis is not appropriate here. The information provided here can be included in the discussion but not discussed as sensitivity analysis”

Suggestion accepted. We discussed this analysis in the proper section. Lines 265-274.

“add " to the best of our knowledge"” 

“Remove the word reliable as there is no way your audience can confirm that adjective. It is ok to write that you provided data on resources used but you don't need to stress the reliability. The reliability of resources may be difficult to measure in this context.”

Suggestions accepted. Lines 231-233

“If you say numerous studies, you should cite references backing that statement.”

Suggestion accepted. Lines 246-247

“Rather say the result may not be generalized since it is a limitation and not that it can be generalized.”

Suggestion accepted. We have rephrased the sentence to improve comprehension. Lines 291-293.

---

## [Decision Letter · Decision Letter 1]

6 Oct 2022

PONE-D-22-19266R1Medical cost of acute diarrhea in children in ambulatory carePLOS ONE

Dear Dr. Jimbo,

Thank you for submitting your manuscript to PLOS ONE. After careful consideration, we feel that it has merit but does not fully meet PLOS ONE’s publication criteria as it currently stands. Therefore, we invite you to submit a revised version of the manuscript that addresses the points raised during the review process.

We look forward to receiving your revised manuscript.

Kind regards,

Nasser Hadal Alotaibi

Academic Editor

PLOS ONE

Journal Requirements:

Reviewers' comments:

Reviewer's Responses to Questions

**Comments to the Author**

1. If the authors have adequately addressed your comments raised in a previous round of review and you feel that this manuscript is now acceptable for publication, you may indicate that here to bypass the “Comments to the Author” section, enter your conflict of interest statement in the “Confidential to Editor” section, and submit your "Accept" recommendation.

Reviewer #1: All comments have been addressed

Reviewer #2: All comments have been addressed

2. Is the manuscript technically sound, and do the data support the conclusions?

Reviewer #1: Partly

Reviewer #2: Yes

3. Has the statistical analysis been performed appropriately and rigorously? 

Reviewer #1: No

Reviewer #2: Yes

4. Have the authors made all data underlying the findings in their manuscript fully available?

Reviewer #1: Yes

Reviewer #2: Yes

5. Is the manuscript presented in an intelligible fashion and written in standard English?

Reviewer #1: Yes

Reviewer #2: Yes

6. Review Comments to the Author

Reviewer #1: The manuscript is improved as compared to last version. Authors have done a good job. English is highly improved.

Reviewer #2: I think the authors have done justice to the earlier queries. However, the conclusion needs a slight modification. Rather than providing a need for more research, it might be nice to offer recommendation to mitigate the challenges identified from the study in the conclusion. I know that some recommendations were in the body of the discussion, but it would be nice to input a summarized version of the recommendation in the conclusion in a concise manner.

7. PLOS authors have the option to publish the peer review history of their article (what does this mean?). If published, this will include your full peer review and any attached files.

Reviewer #1: No

Reviewer #2: **Yes: **Oluwasola Stephen Ayosanmi

---

## [Author Response · Author response to Decision Letter 1]

17 Oct 2022

Point by point response letter

Journal requirements

Corrections were made to meet PLOS ONE’s journal requirements. Reference number 12 was changed. Brownlee, 2017. (With an erratum) by Elshaug, 2017. Line 340.

Reviewer #1: 

“The manuscript is improved as compared to last version. Authors have done a good job. English is highly improved.”

No changes were made.

Reviewer #2: 

“I think the authors have done justice to the earlier queries. However, the conclusion needs a slight modification. Rather than providing a need for more research, it might be nice to offer recommendation to mitigate the challenges identified from the study in the conclusion. I know that some recommendations were in the body of the discussion, but it would be nice to input a summarized version of the recommendation in the conclusion in a concise manner.

Corrections were made in the conclusion section. Lines 300-304.

---

## [Decision Letter · Decision Letter 2]

5 Dec 2022

Medical cost of acute diarrhea in children in ambulatory care

PONE-D-22-19266R2

Dear Dr. Jimbo,

We’re pleased to inform you that your manuscript has been judged scientifically suitable for publication and will be formally accepted for publication once it meets all outstanding technical requirements.

Kind regards,

Nasser Hadal Alotaibi

Academic Editor

PLOS ONE

Additional Editor Comments (optional):

Reviewers' comments:

Reviewer's Responses to Questions

**Comments to the Author**

1. If the authors have adequately addressed your comments raised in a previous round of review and you feel that this manuscript is now acceptable for publication, you may indicate that here to bypass the “Comments to the Author” section, enter your conflict of interest statement in the “Confidential to Editor” section, and submit your "Accept" recommendation.

Reviewer #2: All comments have been addressed

2. Is the manuscript technically sound, and do the data support the conclusions?

Reviewer #2: Yes

3. Has the statistical analysis been performed appropriately and rigorously? 

Reviewer #2: I Don't Know

4. Have the authors made all data underlying the findings in their manuscript fully available?

Reviewer #2: Yes

5. Is the manuscript presented in an intelligible fashion and written in standard English?

Reviewer #2: Yes

6. Review Comments to the Author

Reviewer #2: The author was asked to improve on the concluding part of the manuscript. The author has addressed the concerns highlighted in the initial reviews

7. PLOS authors have the option to publish the peer review history of their article (what does this mean?). If published, this will include your full peer review and any attached files.

Reviewer #2: **Yes: **Ayosanmi Oluwasola Stephen

---

## [Editor Report · Acceptance letter]

8 Dec 2022

PONE-D-22-19266R2 

Medical cost of acute diarrhea in children in ambulatory care 

Dear Dr. Jimbo:

I'm pleased to inform you that your manuscript has been deemed suitable for publication in PLOS ONE. Congratulations! Your manuscript is now with our production department. 

Kind regards, 

on behalf of

Dr. Nasser Hadal Alotaibi 

Academic Editor

PLOS ONE